# Different Microeukaryotic Trophic Groups Show Different Latitudinal Spatial Scale Dependences in Assembly Processes across the Continental Shelves of China

**DOI:** 10.3390/microorganisms12010124

**Published:** 2024-01-08

**Authors:** Yong Zhang, Zhishuai Qu, Kexin Zhang, Jiqiu Li, Xiaofeng Lin

**Affiliations:** 1Key Laboratory of Ministry of Education for Coastal and Wetland Ecosystems, Fujian Province Key Laboratory for Coastal Ecology and Environmental Studies, College of the Environment and Ecology, Xiamen University, Xiamen 361102, China; zhangyong_aqr@126.com (Y.Z.); zqu@xmu.edu.cn (Z.Q.); zhangkexin9505@163.com (K.Z.); lijiqiu@xmu.edu.cn (J.L.); 2School of Health and Life Sciences, University of Health and Rehabilitation Sciences, Qingdao 266071, China

**Keywords:** deterministic processes, microeukaryotes, sediment, stochastic processes, surface water

## Abstract

The relative role of stochasticity versus determinism is critically dependent on the spatial scale over which communities are studied. However, only a few studies have attempted to reveal how spatial scales influence the balance of different assembly processes. In this study, we investigated the latitudinal spatial scale dependences in assembly processes of microeukaryotic communities in surface water and sediment along the continental shelves of China. It was hypothesized that different microeukaryotic trophic groups (i.e., autotroph, heterotroph, mixotroph, and parasite) showed different latitudinal scale dependences in their assembly processes. Our results disclosed that the relative importance of different assembly processes depended on a latitudinal space scale for planktonic microeukaryotes. In surface water, as latitudinal difference increased, the relative contributions of homogenous selection and homogenizing dispersal decreased for the entire community, while those of heterogeneous selection and drift increased. The planktonic autotrophic and heterotrophic groups shifted from stochasticity-dominated processes to heterogeneous selection as latitudinal differences surpassed thresholds of 8° and 16°, respectively. For mixotrophic and parasitic groups, however, the assembly processes were always dominated by drift across different spatial scales. The balance of different assembly processes for the autotrophic group was mainly driven by temperature, whereas that of the heterotrophic group was driven by salinity and geographical distance. In sediment, neither the entire microeukaryotic community nor the four trophic groups showed remarkable spatial scale dependences in assembly processes; they were always overwhelmingly dominated by the drift. This work provides a deeper understanding of the distribution mechanisms of microeukaryotes along the continental shelves of China from the perspective of trophic groups.

## 1. Introduction

Comprehensive understanding of biodiversity and geographic patterns of microorganisms has been significantly improved via advances in high–throughput sequencing [1]. Increasing evidence was proved to document that microbial similarity decreases within increasing spatial distance, following the distance–decay pattern [2]. By comparison, the ecological processes driving the distribution patterns are not well understood [3]. Ecological processes frequently operate within a metacommunity context, which enables the evaluation of community assembly at multiple scales [4]. According to the framework proposed by Vellend [5], ecological processes shaping community assemblages can be classified into four fundamental types: selection, dispersal, ecological drift, and speciation. Selection can act in opposite directions, either reducing diversity through homogeneous selection or increasing diversity within communities through heterogeneous selection, depending on biotic or abiotic conditions. These types of selection are deterministic processes. Dispersal is the movement of organisms across space, with a low dispersal rate (dispersal limitation) or a high dispersal rate (homogenizing dispersal). Dispersal processes are typically considered as stochastic. Ecological drift (i.e., drift) is an absolute stochastic process, that refers to the effect derived from stochastic birth, death, offspring production, immigration, and emigration. Speciation, the evolution of new species, is not considered due to its relatively limited impact on the metacommunities and the considerable challenges associated with measurement in natural environments [6,7,8]. Recently, unraveling the relative contribution of deterministic and stochastic processes could enable significant progress in understanding the community assembly [9]. However, previous studies have reported that the relative contribution of each process varies across different spatiotemporal scales [10,11,12]. Therefore, a crucial unresolved issue is how spatial idiosyncrasies influence the interpretation of mechanisms governing microbial community assembly.

Microeukaryotes, characterized by their vast taxonomic diversity and diverse trophic modes, play key functional roles in both the first several trophic links in food webs and the cycling of biogeochemical elements [13,14]. For instance, autotrophs form the foundation of the marine food chains, facilitating energy transfer, resource provision, and recycling for higher trophic levels within the ecosystem [14,15]. Heterotrophs and mixotrophs contribute significantly to the remineralization of carbon and nutrients derived from primary production through the microbial loop [16]. Ecological studies using different trophic groups can simultaneously help to provide a more comprehensive acknowledgement of the underlying mechanisms shaping the ocean microbiota at different evolutionary levels [17]. Interactions among trophic groups vary along the latitude gradient and respond differently to global warming [18,19]. The distinct latitudinal patterns evident in both extant diversity and future changes in diversity, community turnover rate, and size structure were revealed using global marine model estimations [20,21]. It indicates that the relative contributions of assembly processes governing microeukaryotic communities could exhibit scale-dependent variations in latitude gradients. A previous study focused on three taxonomic groups has shown the impact of latitudinal variation in assembly processes on microeukaryotic communities [12]. However, the latitudinal scale dependences of assembly processes on groups with distinct trophic traits are largely unresolved.

Continental shelf ecosystems, encompassing the water column and sediment, harbor a rich diversity of microeukaryotic plankton and serve as hotspots for global warming impacts [22,23]. The continental shelves of China span three ecoregions: the Cold Temperate Northwest Pacific, the Warm Temperate Northwest Pacific, and the South China Sea [24,25]. They exhibit distinct characteristics, including continuous latitude gradients and heterogeneous environmental conditions [26]. Thus, these ecosystems provide ideal opportunities for studying the balance of different assembly processes in latitude gradients. We hypothesized that the spatial scale dependence of assembly processes in microeukaryotic communities may be different for various trophic groups. Therefore, we carried out surveys along a latitudinal gradient spanning from 19° N to 40° N on the continental shelves of China. The eDNA metabarcoding method was employed to explore the assembly processes shaping the geographic patterns of microeukaryotic communities and their latitudinal scale dependences in four trophic groups. Our main objectives were as follows: (1) To determine whether assembly processes in planktonic and sedimental habitats exhibit same variation tendencies along latitudinal scales; (2) To investigate how the relative importance of assembly processes in different trophic groups varies along latitudinal scales; (3) To identify the key drivers controlling the balance of community assembly processes.

## 2. Materials and Methods

### 2.1. Study Area and Sampling

Sampling was carried out on the continental shelves of China. The sampling sites were determined along the coastline of China, spinning approximately from 111° to 126° E longitude and 19° to 40° N latitude. These sites covered three distinct ecoregions: the Cold Temperate Northwest Pacific (ecoregion I), the Warm Temperate Northwest Pacific (ecoregion II), and the South China Sea (ecoregion III) (Figure 1A,B) [25]. A total of forty-four sites were sampled during three cruises conducted between March and June 2018. A total of 2 L of surface water (0–6 m) was collected using Niskin bottles at each of the forty-four sites (Figure 1A). The collected water was filtered using a 0.65 μm pore sized polycarbonate membrane with a diameter of 47 mm (Millipore, Billerica, MA, USA). The surface sediment (0–5 cm) samples were obtained from twenty-five of these sites (Figure 1B) using a sediment sampler and subsequently transferred into sterile aluminum containers. All of the sixty-nine collected samples were promptly frozen using liquid nitrogen and thereafter stored at a temperature of −80 °C until further processing. During the sampling process, the equipped CTD (conductivity, temperature, and depth) sensor was employed to log the salinity, temperature, and depth, with details presented in Appendix A.

### 2.2. Molecular Analyses

The DNA was extracted from the water sample membranes and sediment samples (0.3 g for each) using a PowerSoil^®^ DNA Isolation Kit (MOBIO, Carlsbad, CA, USA) following the manufacturer’s protocol. The hypervariable V4 region of the eukaryotic 18S rRNA gene was amplified using the specific primer 528F and 706R [27]. The PCR amplification was performed using Phusion^®^ High-Fidelity PCR Master Mix with GC Buffer (New England Biolabs Inc., Beverly, MA, USA). Subsequently, the sequencing libraries were generated using a TruSeq^®^ DNA PCR-Free Sample Preparation Kit. Following library preparation, sequencing was executed on the Illumina NovaSeq platform (Illumina Inc., San Diego, CA, USA). Subsequent to sequencing, a series of data-processing steps were undertaken. The raw tags were subjected to quality filtering using QIIME v.1.9 [28], based on the criteria of truncation of tags at the site preceding three consecutive bases of low quality. Furthermore, tags in which the fraction of consecutive high-quality bases fell below 75% of the total tag length were eliminated [29]. Chimeras were identified and removed using the UCHIME algorithm [30]. The remaining high-quality, clean tags were then clustered into OTUs via UPARSE v.7.0 [31], with a defined threshold of 97% similarity.

### 2.3. Taxonomic and Trophic Compositions

The taxonomic assignment of each OTU was generated using BLAST within the QIIME program against the Protist Ribosomal Reference (PR2) database [32]. For data analysis, OTUs not belonging to the target taxonomic groups (including metazoans, embryophytes, and other multicellular organisms), unassigned OTUs, and OTUs containing fewer than two tags were excluded. Furthermore, a randomly subsampled subset (cutoff = 18,244) was extracted from each sample to the standardize sequencing effort across the samples. Finally, these taxonomic taxa were classified into four trophic groups, autotroph, heterotroph, mixotroph, and parasite, primarily based on information from the literature [33,34,35]. Unresolved taxa and unnamed species that could not be attributed to any specific trophic group were annotated as “Unknown”. Sub-datasets (a total of sixty-seven samples) for each trophic group were extracted from the original OTU table and normalized through random subsampling. The samples B22.w and E11.s were excluded from the trophic-group-based analyses due to their extremely low abundance of the mixotrophic group.

### 2.4. Analysis of Microeukaryotic Community

Alpha diversity was assessed using the Shannon diversity index. The significant differences in alpha diversity between sample groups were compared using the Wilcoxon rank-sum test. Beta diversity was calculated using Bray–Curtis distances matrices. Non-metric multidimensional scaling (NMDS) was performed to visualize community dissimilarities based on the Bray–Curtis distances. The significance of the community compositional changes among sample groups was demonstrated using an analysis of similarities (ANOSIM). To explore the distance–decay patterns of microeukaryotic communities across latitudinal gradients, we examined the relationships between community similarities (measured using Bray–Curtis distances) and pairwise latitudinal differences for all members and the four trophic groups of microeukaryotes. These relationships were tested using Spearman’s rank correlations. The slopes of distance–decay curves were calculated through linear least-squares regression, which related community similarities and latitudinal differences. Most of the statistical analyses described above were conducted using the “vegan” package [36] in R version 4.2.0 [37].

### 2.5. Ecological Processes Analyses

The null model, based on the framework developed by Stegan et al. [8], was employed to estimate the contribution of different ecological processes to community assembly. Ecological processes analyses were conducted using the R package “picante” [38]. First, the abundance-weighed β-mean nearest taxon distance (βMNTD) metric was calculated, measuring the mean phylogenetic distances between the two evolutionarily closest OTUs within two communities. Second, the β-nearest taxon (βNTI) metric quantifies the magnitude and direction of the deviation of the observed βMNTD from the null distribution of βMNTD. It provides an estimation of phylogenetic turnover, accounting for stochastic and deterministic ecological processes [39]. βNTI values > 2 indicate heterogeneous selection, meaning a significantly greater phylogenetic turnover than expected. Conversely, βNTI values < −2 indicate homogeneous selection, indicating significantly less phylogenetic turnover than expected. When |βNTI| > 2 (with no significant deviation from the null βMNTD distribution), this indicates stochastic processes. A subsequent step involved the assessment of whether community structure could be shaped by dispersal or drift, utilizing the Bray–Curtis based Raup–Crick metric (RC_bray_). Values of RC_bray_ > 0.95, |RC_bray_| < 0.95, and RC_bray_ < −0.95 are interpreted as indicative of dispersal limitation, drift, and homogenizing dispersal, respectively. Finally, the relative contribution of each ecological process to community assembly was expressed as a percentage in all sample pairs.

In order to assess the variation in community assembly processes across latitudinal gradients, regression analyses were employed to compare the βNTI values with latitudinal difference matrices. Considering the correlation between latitude and factors such as temperature, geographical distance, and salinity, we performed standard and partial Mantel tests to examine the relationships between these factors and βNTI values. These tests enabled the evaluation of key factors that mediate the relative contributions of ecological processes in community assembly across latitudinal scales.

## 3. Results

### 3.1. Microeukaryotic Community Composition

After quality filtering, the sequencing of sixty-nine samples resulted in 1,258,836 high-quality tags assigned to 8438 OTUs, belonging to 33 phyla (Appendix A). Analysis of the Shannon index indicated that the alpha diversity showed no significant difference between sedimental and planktonic habitats (*p* = 0.088) (Figure 1C).

Taxonomic groups exhibited distinct distributions and compositions across habitats and ecoregions. In terms of the number of tags, the phylum Dinophyta was the most abundant group in both the planktonic and sedimental habitats (57.7% and 39.1%, respectively), as well as across the three ecoregions (31.8–66.4%). The phylum Cercozoa exhibited greater abundance in sedimental habitats (15.6% in all samples, 16.0% in ecoregion I, and 15.2% in ecoregion II) in comparison to planktonic habitats (2.9% in all samples; 5.9%, 1.2%, and 1.0% in ecoregions I, II, and III, respectively). In terms of OTUs, Dinophyta exhibited the highest richness in planktonic habitats (39.0% in all samples; 28.8%, 40.5%, and 48.4% in ecoregions I, II, and III, respectively), whereas Cercozoa dominated the sedimental habitats (29.2% in all samples; 32.2% and 28.1% in ecoregions I and II, respectively) (Figure 2A).

The distribution and composition of trophic groups were similar across habitats and ecoregions, with the heterotrophic group predominating. In the planktonic habitat, 40.4% of OTUs and 44.9% of tags were affiliated with heterotrophs, while in the sedimental habitat, 53.3% of OTUs and 45.3% of tags were affiliated with heterotrophs. Parasites constituted the second richest trophic group, exhibiting high relative abundances in both habitats. They accounted for 30.9% of OTUs and 18.4% of tags in the surface water, and 25.6% of OTUs and 12.3% of tags in the sediment overall (Figure 2B).

In terms of the taxonomic composition within trophic groups, the majority of autotrophs were attributed to Ochrophyta (76.6% of OTUs in planktonic autotrophs; 74.2% of OTUs in sedimental autotrophs). Almost half of the mixotrophs were affiliated with Dinophyta (42.6% of OTUs in planktonic mixotrophs; 49.8% of OTUs in sedimental mixotrophs). Cercozoa and Ciliophora were the major components of heterotrophic group in both planktonic (31.3% of OTUs for Cercozoa; 17.7% of OTUs for Ciliophora) and sedimental habitats (42.6% of OTUs for Cercozoa; 12.9% of OTUs for Ciliophora). However, there were noticeable differences in the composition of parasitic groups between the two habitats. The parasitic group in planktonic systems was mainly constituted of Dinophyta, which exhibited absolute dominance (79.3% of OTUs in planktonic parasites). In contrast, the parasitic group in sedimental systems primarily consisted of Dinophyta (34.7% of OTUs), Apicomplexa (21.6% of OTUs), and Cercozoa (17.8% of OTUs sedimental) (Figure 2C).

### 3.2. Spatial Distribution Patterns of Microeukaryotic Communities

The NMDS analysis revealed distinct spatial distributions of the microeukaryotic communities within the continental shelf system. These communities exhibited separation according to habitat types and ecoregional delineations. First, the ecosystem boundary between water and sediment in the continental shelf system was validated for the entire microeukaryotic communities (*p* < 0.01 in the global ANOSIM test). Second, in the surface water and sediment, microeukaryotic communities exhibited distinct community separation based on ecoregions, accompanied by latitudinal variation (both *p* < 0.01 in the global ANOSIM test) (Figure 3A and Appendix A). Furthermore, this ecoregional distribution pattern was observed in all four trophic groups, except for the mixotrophic group in sediment. The sedimental mixotrophic group showed no significant ecoregional distribution pattern (*p* = 0.125 in the global ANOSIM test) (Appendix A).

We further assessed the latitudinal distance–decay patterns of microeukaryotes within the continental shelf system. The results demonstrated a significant decline in community similarities among microeukaryotes with increasing latitudinal differences (*p* < 0.01 in Spearman test). Moreover, for both the entire community and different trophic groups, the distance–decay rates of planktonic microeukaryotes (slope: entire community, −0.023; autotrophs, −0.021; heterotrophs, −0.016; mixotrophs, −0.021; and parasites, −0.021) were steeper than those of sedimental microeukaryotes (slope: entire community, −0.014; autotrophs, −0.018; heterotrophs, −0.010; mixotrophs, −0.012; and parasites, −0.015). In the same habitat, the distance–decay patterns of the four trophic groups, i.e., autotrophs, heterotrophs, mixotrophs, and parasites, demonstrated the same trend (Figure 3B,C). Overall, similar spatial distribution patterns of microeukaryotes were observed across different habitats and different trophic groups.

### 3.3. Ecological Processes Governing Microeukaryotic Communities

There were certain differences in the assembly of microeukaryotic communities between surface water and sediment on the continental shelves. Specifically, while the drift contributed the largest fraction to the assembly of both planktonic (56.7%) and sedimental (73.7%) communities overall, homogenous selection and homogenizing dispersal contributed more to community assembly in the planktonic communities as a whole (19.6% and 14.9%, respectively), compared to the sedimental communities as a whole (1.3% and 3.0%, respectively). Conversely, heterogeneous selection exerted a stronger impact on the assembly of entire sedimental communities (22.0%) than that of entire planktonic communities (8.9%) (Figure 3D).

There were variations in assembly processes observed among different trophic groups within the same habitat. In the surface water, apart from the dominant processes of drift, heterogeneous selection had a greater contribution for autotrophic (44.2%) and heterotrophic groups (22.1%), and homogenizing dispersal had a smaller contribution for these two communities (2.4% for autotrophs and 8.7% for heterotrophs). However, homogenizing dispersal (18.1%) had a greater contribution to the mixotrophic group compared to the other trophic groups. The homogenous selection (18.5%) and homogenizing dispersal (17.7%) had comparable contributions for the parasitic group, similar to the contributions of assembly processes in the entire planktonic community. In the sedimental habitat, homogenous selection (15.9%) had a greater influence on the heterotrophic group, and homogenizing dispersal (10.9%) had a larger contribution to the mixotrophic group compared to the others. However, heterogeneous selection (18.5%) had a more substantial contribution to the parasitic group, mirroring the pattern observed in the entire sedimental communities (Figure 3D).

### 3.4. Spatial Scale Dependence of Microeukaryotic Community Assembly

We investigated the assembly of microeukaryotic communities across latitudinal scales within the surface water and sediment of the continental shelves. The results revealed that the distribution of βNTI values shifted significantly with increasing latitudinal differences in the planktonic habitat (R^2^ = 0.281, *p* < 0.01; Figure 4A). We manually divided the latitudinal differences into four latitudinal gradients, i.e., 0–5°, 5–10°, 10–15°, and 15–21°, to present a concrete representation of the changes in various assembly processes. The relative contributions of homogenous selection decreased from 36.8% to 1.2% and homogenizing dispersal decreased from 20.7% to 1.2%, while those of heterogeneous selection increased from 1.7% to 16.1% and drift increased from 40.8% to 81.3% with increasing latitudinal differences (Figure 4B). However, the distribution of βNTI along the latitudinal gradients in sedimental communities exhibited slight shifts with weak support (R^2^ = 0.014, *p* < 0.05). The distribution of βNTI values largely fell within the region of drift (69.3–81.1%) (Figure 4A,B).

In the surface water, the relationships between βNTI values and latitudinal differences varied among the four trophic groups. This indicated that the increasing differences in latitude led to a shift from stochasticity to heterogeneous selection in the community assembly of autotrophs (R^2^ = 0.252, *p* < 0.01) and heterotrophs (R^2^ = 0.401, *p* < 0.01). However, the assembly processes of mixotrophic (R^2^ = 0, *p* > 0.05) and parasitic groups (R^2^ = 0.024, *p* < 0.01) were dominated by stochasticity throughout the latitudinal gradient. In the sedimental habitat, the distribution of βNTI values in either trophic group predominantly fell within the region of stochasticity across different spatial scales (Figure 4C). Further analysis was conducted on the changes in the dominant assembly processes of planktonic autotrophic and heterotrophic groups by categorizing latitudinal differences using finer gradients. This revealed that the transition of dominant processes in the planktonic autotrophic group occurred at a smaller spatial threshold scale. In contrast, the same transition in the planktonic heterotrophic group occurred at a larger spatial threshold scale. For the planktonic autotrophic group, stochasticity played a dominant role for latitudinal differences less than 8°, and heterogeneous selection dominated for differences greater than 8°. For the planktonic heterotrophic group, stochasticity played a dominant role for latitudinal differences less than 16°, and heterogeneous selection dominated for differences greater than that (Appendix A).

After controlling for other factors using a partial Mantel test, we found that temperature was the most important factor in determining the variation of βNTI values in the planktonic microeukaryotic communities as a whole (*r* = 0.216, *p* < 0.01), followed by salinity with weaker correlations (*r* = 0.151, *p* < 0.01). Latitude and geographical distance did not exhibit any significant correlations (*p* > 0.05). For different trophic groups in the surface water, temperature was identified as the predominant driver of βNTI variation in autotrophic (*r* = 0.562, *p* < 0.01) and parasitic groups (*r* = 0.145, *p* < 0.05). In contrast, βNTI values in the heterotrophic group were mainly driven by salinity (*r* = 0.207, *p* < 0.01) and geographical distance (*r* = 0.200, *p* < 0.01) and were not significantly correlated with temperature and latitude (*p* > 0.05). No significant correlation between these factors and βNTI values of the mixotrophic group (*p* > 0.05). In sediment, no factor was identified as a driver for the assembly processes of the entire and the four trophic groups (*p* > 0.05) (Appendix A).

## 4. Discussion

### 4.1. Different Taxonomic but Similar Trophic Compositions between Planktonic and Sedimental Habitats

Previous studies found that microeukaryotic communities in sediment were more diverse than those in surface water [40,41]. The characteristics of sediment, such as higher temporal stability, greater diversity of niches, and resource partitioning, are widely recognized as factors that promote species diversification and coexistence [42,43]. Additionally, the migration of surface species to deep waters through passive or active vertical flux may also contribute to the higher diversity values in the sediment [44]. This study, along with others, confirmed the significant differences in the taxonomic diversity of microeukaryotes between planktonic and sedimental habitats [45,46]. By richness, the planktonic habitats are dominated by the phylum Dinophyta, while the sedimental habitats are dominated by Cercozoa [47,48]. The majority of Dinophyta members live in marine environments and mainly exist as planktonic organisms, with benthic groups accounting for only 8% [49]. Cercozoa are a rare and low-diversity marine group and appear to be dominant in soils [50]. This suggested that microbial webs within sedimental environments differ from plankton in marine ecosystems [48].

In spite of the observed differences in taxonomic compositions between planktonic and sedimental habitats, the compositions of trophic groups were similar in that both were dominated by heterotrophic groups. It is worth noting that the heterotrophic groups are not only dominant in marine ecosystems but also in freshwater and soil [51,52]. The heterotrophic strategy has been demonstrated to be the over-dominant nutritional mode of protists, and these organisms play a crucial role in regulating the population dynamics and community assembly of both bacterial and eukaryotic preys [53,54]. Furthermore, the results also indicated a remarkable functional homogeneity among microeukaryotic communities of various habitats [55]. In our study, this was mainly due to the differences in the taxonomic composition of parasitic groups in the two habitats. Parasitic groups were overwhelmingly dominated by Dinophyta in planktonic systems, whereas in sedimental systems they were constituted by comparable diversity of Dinophyta, Apicomplexa, Cercozoa, etc.

### 4.2. Similar Biogeographical Patterns with Distinct Assembly Mechanisms

Increasing evidence has shed light on the limited geographical distribution of microorganisms, in contrast to their presumed global distribution [56]. In our study, similar biogeographical patterns of microeukaryotic communities were observed in both surface water and sediment, and these patterns aligned with the ecoregional delineations. The marine ecoregion is a global bioregionalization of coastal and shelf areas and is used widely in macroorganisms [25]. Recent studies on sedimental archaea and ciliates, and planktonic microeukaryotes also showed that the biogeographical patterns of microorganisms, similar to macroorganisms, correspond well with the ecoregional delineations [26,55,57]. Our findings confirmed that the clustering of communities was primarily driven by the habitat types (e.g., water vs. sediment) and, within each habitat type, there were notable regional effects (e.g., ecoregions) [58]. In addition, accompanying the geographic variations were the distance–decay relationships in all microeukaryotic communities, with plankton exhibiting at steeper rates than benthos. This disparity suggested the disparate dispersal potentials of planktonic and benthic microorganisms [48]. Overall, planktonic organisms exhibit stronger dispersal abilities compared to benthic organisms [59]. In addition, sediment represents a more heterogeneous environment compared to the water column, which limits the opportunities for active dispersal [26,60]. Therefore, greater community dissimilarity in microeukaryotes was observed among neighboring benthic sites [61].

In our study, the biogeographical patterns of microeukaryotic communities in planktonic and sedimental habitats were driven by different contributions of assembly processes. In the context of metacommunity, the strength of the distance–decay relationship is influenced by both deterministic and stochastic processes [4,62], whereas the relative contributions depend on the microbial groups under study [12,63]. We found that both planktonic and sedimental communities were dominantly shaped by drift, which is considered to be the main assembly mechanism in a metacommunity [17]. However, in the community assembly of planktonic microeukaryotes, homogenous selection and homogenizing dispersal played more significant roles, while sedimental communities were more associated with heterogeneous selection on the whole. The prevalence of homogenous selection has also been found in planktonic prokaryotic communities [17], suggesting that several consistent environmental factors select similar microbial communities locally [26]. Due to the greater stability and reduced influence of tidal mixing within sediments, sedimental communities exhibit stronger spatial heterogeneity, leading to stronger heterogeneous selection on the community assembly [61,64]. In addition, sedimental communities were influenced more by drift than planktonic communities. Drift tends to weaken the distance–decay pattern by homogenizing the communities [65], which might explain the lower distance–decay rates of microeukaryotic communities observed in the sediment compared to those in the water column in our study. The lower dispersal abilities of sedimental microorganisms might also enhance the effect of stochasticity on the community assembly within sediment [26].

Interestingly, we uncovered the trophic group specificity of microbial community assembly, which was a previously underemphasized factor for explaining the balance between deterministic and stochastic processes [66]. For instance, in the planktonic habitat, autotrophic and heterotrophic groups were structured mainly by heterogeneous selection, indicating their high sensitivity to environmental changes [62]. Conversely, the mixotrophic group exhibited a stronger homogenizing dispersal, implying a higher spatial dispersal rate compared to the other trophic groups [6]. The assembly processes of the parasitic group in both planktonic and sedimental habitats was found to be similar to those of entire communities, supporting the hypothesis that parasitic groups may play a crucial role in the microbial food webs [67]. This finding suggested that various trophic groups within microbial communities respond differently to the same environmental conditions in the oceans. To comprehend the response of the entire microbial food web to global change, it is essential to consider the traits of these assemblages. However, the specific roles played by individual groups in the assembly of the entire microbial community remain poorly understood [12,26]. Further investigation is needed from various perspectives, including, but not limited to, evolutionary rates [26], ecological characteristics [58], and biological interactions [17].

### 4.3. Latitudinal Spatial Scale Dependence of Assembly Processes in Planktonic Communities Contrasting Sedimental Communities

Our data revealed latitudinal scale variations of different assembly processes within microeukaryotic communities in the planktonic habitat. At smaller scales, the observed trends in these processes indicated that the high microbial exchange rate led to stronger homogenous dispersal, and the presence of similar environments resulted in weaker heterogeneous selection [68]. As spatial scale increased, several factors contributed to the greater effects of dispersal limitation and heterogeneous selection for microorganisms. These factors included the failure of microbial establishment, increased environmental heterogeneity, and the influence of different currents [9,17,69]. Surprisingly, we found that the variation in assembly processes within planktonic communities was primarily influenced by temperature rather than latitude and geographical distance. This finding was consistent with previous marine-based research, which suggested that temperature appeared to dominate the assembly processes of planktonic communities [70,71]. However, this contrasted with soil-based studies that consider geographic distance as the significant influencing factor [11,72]. These results highlighted that, in the future, epipelagic microeukaryotic diversity and ecosystem services may receive strong impacts mainly from rising temperatures, as previously predicted for planktonic communities [21,73].

Nonetheless, the community assembly of sedimental microeukaryotes showed no remarkable latitudinal scale-dependent shifts and exhibited no significant correlations with the measured factors here. One possible reason was that the sedimental system comprises more discrete habitats, where abiotic factors display a patchy distribution even at small scales [54]. In contrast to planktonic communities, the influence of temperature on sedimental communities is limited [21,44], leading to a lower level of selection for the latter. Consequently, at increasing spatial scales, other unmeasured factors (referred as drifts) consistently emerged as the primary drivers of community assembly in sedimental habitats. Another potential explanation could be that the spatial range of the sedimental samples covered in the present study (approximately 1487 km) might not have been sufficient to fully capture the dynamics of the balance between assembly processes in sedimental communities. Previous observations based on soil bacterial communities have manifested that stochasticity plays a dominant role up to 900 km, while determinism becomes prevalent beyond this distance. This change is primarily driven by a substantial pH gradient across the geographic range [11]. In contrast, Liu et al. [26] showed that stochasticity, rather than determinism, has a more pronounced impact on the archaeal community assembly across approximately 1500 km in the eastern Chinese marginal seas. Therefore, it became evident that microbial communities in sedimental environments are generally less predicted by spatial distance compared to those in water columns [50].

It was noteworthy that the relative importance of stochasticity and determinism in planktonic autotrophic and heterotrophic groups varies according to latitudinal difference, with determinism dominating at larger scales. A previous study has provided evidence that high temperature divergence can drive the community assembly towards being more deterministic, with a prevalence of heterogeneous selection in *Synechococcus* [12]. As mentioned above, temperature emerged as the predominant factor influencing the assembly processes of planktonic microeukaryotic communities. Our findings supported this observation for autotrophic groups in the planktonic habitat, but not for heterotrophic groups. The transition from stochasticity to heterogeneous selection in the assembly of the planktonic heterotrophic group was associated with salinity and geographical distance. This was consistent with the findings of another study, where salinity stress was identified as a driving factor shifting the assembly processes of the microeukaryotic community from stochastic to deterministic [74]. The well-established correlation between temperature and primary producers is widely recognized. The influence of temperature on autotrophic protists is more pronounced compared to larger protists [18,50]. However, heterotrophs were found to be dominant in the aphotic layer, and their community compositions are strongly affected by organic matter availability [21]. These results contribute trophic-group-specific insights to the global study of community assembly in planktonic microorganisms under climatic changes.

Although the trends in the dominant processes that varied across space were similar in planktonic autotrophic and heterotrophic groups, the spatial threshold scales for shifts differed. The spatial scale at which stochasticity is overtaken by determinism may vary depending on the environment gradient and the dispersal ability of species [75]. If environmental selection is weak and species have high dispersal ability, the threshold scale is expected to be larger. Conversely, if environmental selection is strong and species have poor dispersal ability, it is expected to be smaller [72]. The finding that planktonic autotrophs were more strongly driven by environmental selection (experiencing greater heterogeneous selection), while heterotrophic groups exhibited stronger dispersal abilities (displaying greater homogenizing dispersal), might provide an explanation for the larger threshold scale observed in planktonic heterotrophs compared to autotrophs. In summary, in conjunction with the findings of other trophic groups presented in this study, our results indicated that the spatial scale dependences in community assembly varied among different microeukaryotic groups. Specifically, autotrophs displayed the strongest dependence, followed by heterotrophs, parasites, and mixotrophs in the water column. Conversely, all four trophic groups in sediment exhibited the weakest spatial scale dependences for community assembly.

## 5. Conclusions

This study quantitatively discerned the changes in the relative importance of assembly processes in microeukaryotic communities along latitudinal gradients of the continental shelves of China. In surface water and sediment, the geographic distribution of microeukaryotes and four trophic groups exhibited similar patterns, with community assembly predominantly driven by drift. In the planktonic habitat, the relative importance of assembly processes showed significant latitudinal spatial scale dependence in microeukaryotic communities. Temperature emerged as a key factor controlling the balance of different assembly processes in the planktonic microeukaryotic community, providing valuable insights into the potential response of microbial assemblages in marine ecosystems to climate warming. Moreover, the contributions of various assembly processes shaping microbial biogeographical patterns were found to be trophic-group-specific. Our study unveiled the disparities in spatial threshold scales and driving factors that lead to shifts in dominant ecological processes between planktonic autotrophic and heterotrophic groups. In the sedimental habitat, however, this remarkable latitudinal spatial scale dependence was not observed in sedimental microeukaryotic community and trophic groups. Their assembly processes were always dominated by drift across different spatial scales. These findings contribute to a better understanding of the biogeography of microeukaryotic communities from the trophic group perspective.

## Figures and Tables

**Figure 1 microorganisms-12-00124-f001:**
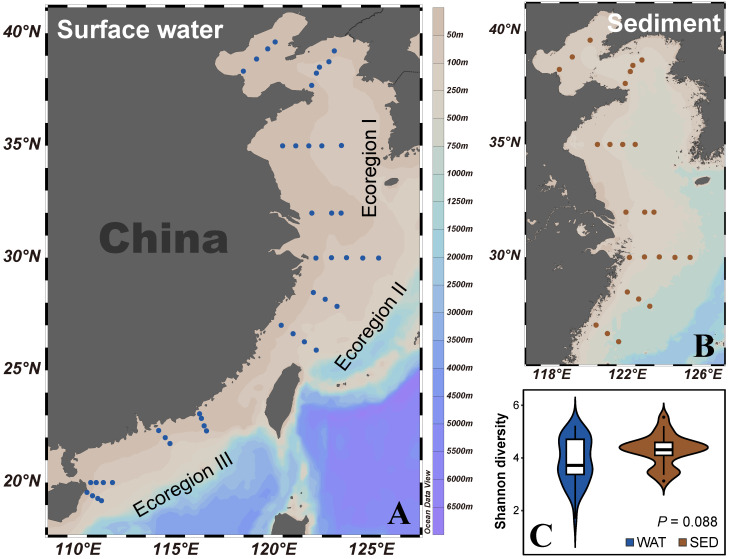
Sampling sites of surface water (**A**) and sediment samples (**B**) on continental shelves of China. (**C**) Comparison of alpha diversity between two habitats. Ecoregions I, II, and III represent the Cold Temperate Northwest Pacific, the Warm Temperate Northwest Pacific, and the South China Sea, respectively. WAT, surface water; SED, sediment.

**Figure 2 microorganisms-12-00124-f002:**
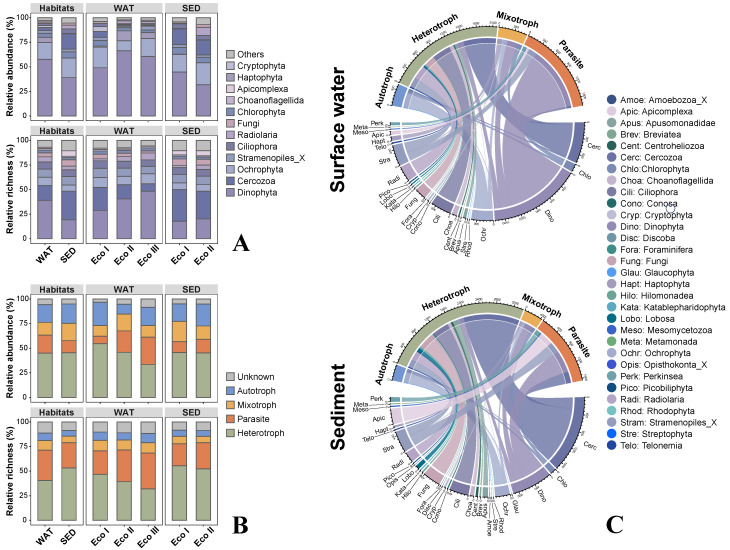
Taxonomic (**A**) and trophic compositions (**B**) of microeukaryotic communities in different sample groups, and the phylum level diversity of the four trophic groups (**C**). Abbreviations for phylum names are given in (**C**). Ecoregions I, II, and III represent the Cold Temperate Northwest Pacific, the Warm Temperate Northwest Pacific, and the South China Sea, respectively. WAT, surface water; SED, sediment.

**Figure 3 microorganisms-12-00124-f003:**
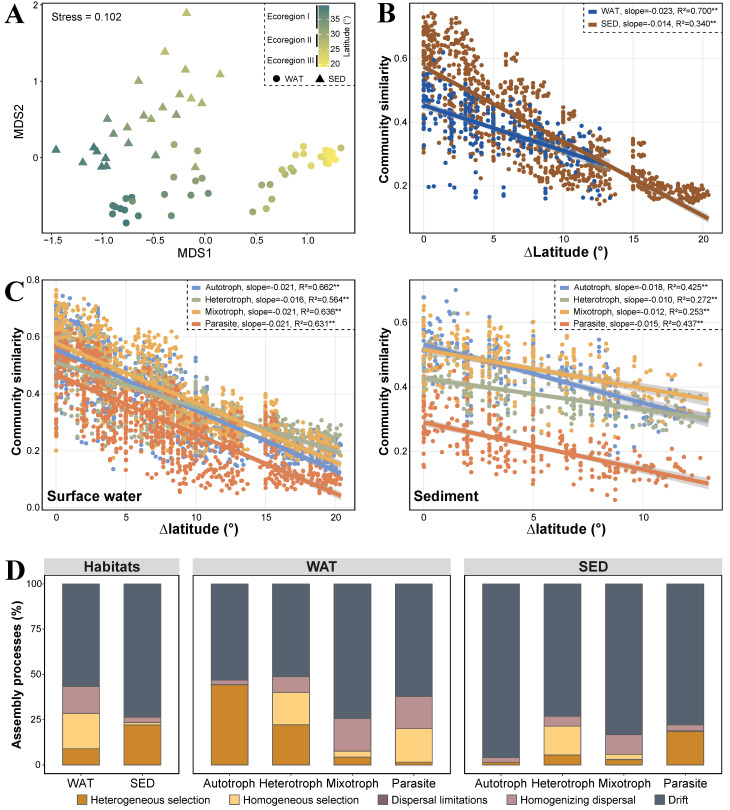
Spatial distribution patterns and assembly processes of microeukaryotic communities. (**A**) Non-metric multidimensional scaling (NMDS) ordination based on the Bray–Curtis distance for all members. Distance–decay patterns based on the community similarity and latitudinal difference for all members (**B**) and the four trophic groups (**C**). Gray shades stand for 95% confidence interval. R^2^ values indicated the correlation coefficient. (**D**) Relative contribution of assembly processes in microeukaryotic communities for all members and the four trophic groups. WAT, surface water; SED, sediment. **, *p* < 0.01.

**Figure 4 microorganisms-12-00124-f004:**
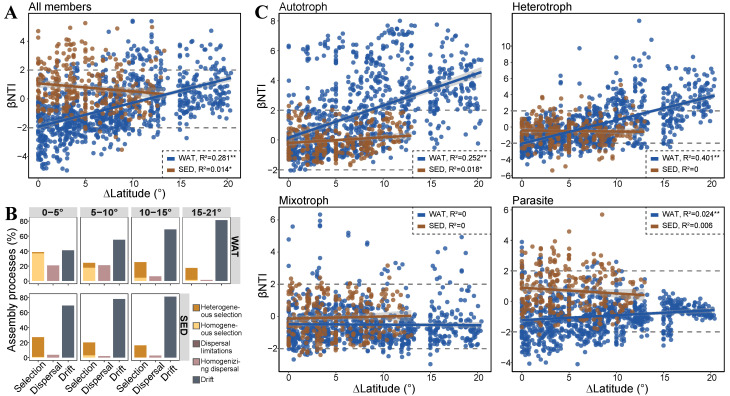
Spatial scale dependences of the assembly processes of microeukaryotic communities. Relationships between βNTI values and latitudinal differences (**A**) and relative contribution of ecological processes grouped by latitudinal difference (**B**) for all members. (**C**) Relationships between βNTI values and latitudinal differences for the four trophic groups. Gray shades stand for 95% confidence interval. R^2^ values indicated the correlation coefficient. WAT, surface water; SED, sediment. **, *p* < 0.01; *, *p* < 0.05.

## Data Availability

Raw sequence data (69 samples) generated in this study have been deposited in the Sequence Read Archive (SRA) database (accessed on 5 September 2023 on https://www.ncbi.nlm.nih.gov/sra/) of the National Center for Biotechnology Information with the project number PRJNA1010514. The data that support the findings of this study are available from the corresponding author upon reasonable request.

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
