# Peer review of "Different Microeukaryotic Trophic Groups Show Different Latitudinal Spatial Scale Dependences in Assembly Processes across the Continental Shelves of China"

_microorganisms, 2024, doi:10.3390/microorganisms12010124_

Round 1

Reviewer 1 Report

Comments and Suggestions for Authors

This paper is interesting and due to its broad spatial scale provides interesting insight into microorganism variation and causal factors in the ocean.

The paper is generally well written.  The text should be carefully reviewed to adjust for English phrasing though.  Prepositions were left out of some sentences.

The paper could be improved by simplifying the text.  Although the wording was correct, reading could be cumbersome due to the extensive use of technical jargon.  More definition of the terminology used in the text and perhaps some simpler phrasing would be helpful. Definitions for repeated phrasing such as alpha diversity, phylogenetic turnover, homogeneous selection, OTU, drift, community assembly processes, and ecological process would be good. 

Simplifying the wording would be particularly useful for the abstract.  Here the text starts right off with technical jargon that might not be clear to many readers. Simplifying (best) or defining terms like stochasticity, determinism or different assembly processes at the beginning would enhance clarity and provide easier identification of the value of the work.

Further, it might be good to give a brief list of the genera in the taxonomic groups in the methods.

 The statistics used were generally sound.  However, please consider enhancing the general numerical descriptions.  For example, it is stated that Dinophyta was the most abundant group across habitats and a percentage was given, but no indication was made if this was a median or mean and no statistical test result was given to demonstrate that it really was greater. 

In cases when the statistical value was stated as not significant, the result cannot be stated as a trend or pattern. Please revise where appropriate.

Please provide the total number of samples for each analysis.  You provided the number of sites.  Were single or multiple samples (replicates?) taken at each site?

The figures were very nice, but the similar colors made it a little difficult to see differences among the groups for Fig 2C. Also Fig. 2C needs more explanation in the legend. Fig. 4 provides an R value in the figure but it is not defined either in the methods or the legend.

The sampling occurred over a short period of time.  It might be interesting to comment on how sampling at a different time or over a longer period might affect the results.

Comments on the Quality of English Language

The English is generally very good.  A careful review is needed to correct for missing prepositions.  

Author Response

This paper is interesting and due to its broad spatial scale provides interesting insight into microorganism variation and causal factors in the ocean.

Response: Thank you for dedicating your time and effort to reviewing our manuscript. Your insightful comments and constructive suggestions have significantly enhanced the quality of our work.

  1. The paper is generally well written. The text should be carefully reviewed to adjust for English phrasing though. Prepositions were left out of some sentences.

Response: We have thoroughly reviewed the English language and rectified errors related to prepositions and others

  1. The paper could be improved by simplifying the text. Although the wording was correct, reading could be cumbersome due to the extensive use of technical jargon. More definition of the terminology used in the text and perhaps some simpler phrasing would be helpful. Definitions for repeated phrasing such as alpha diversity, phylogenetic turnover, homogeneous selection, OTU, drift, community assembly processes, and ecological process would be good. Simplifying the wording would be particularly useful for the abstract. Here the text starts right off with technical jargon that might not be clear to many readers. Simplifying (best) or defining terms like stochasticity, determinism or different assembly processes at the beginning would enhance clarity and provide easier identification of the value of the work.

Response: Thank you for the suggestion. To enhance the readability of the manuscript, we have incorporated a glossary at the end of the paper. This glossary contains definitions for terminology related to assembly processes, including community assembly process, deterministic process, drift, heterogeneous selection, homogeneous selection, homogenizing dispersal, and stochastic process (lines 547-565).

  1. Further, it might be good to give a brief list of the genera in the taxonomic groups in the methods.

Response: A supplementary Table S2, containing a list of OTUs with taxonomic information, including genus assignments, has been provided. Corresponding updates have been implemented in the Results section of the revised manuscript (lines 196).

  1. The statistics used were generally sound. However, please consider enhancing the general numerical descriptions. For example, it is stated that Dinophyta was the most abundant group across habitats and a percentage was given, but no indication was made if this was a median or mean and no statistical test result was given to demonstrate that it really was greater.

Response: The values presented were not median or mean values but rather the relative abundance, expressed as a percentage, for each taxonomic group. For example, the proportion of Dinophyta represents the relative abundance of Dinophyta tags among all OTUs tags in all samples. In studies of this nature, comparing relative abundance is a common method for identifying dominant taxonomic groups based on their ranks, and statistical significance testing is not necessarily applied.

  1. In cases when the statistical value was stated as not significant, the result cannot be stated as a trend or pattern. Please revise where appropriate.

Response: Thank you for the prompt. We have rectified the wording in the revised manuscript. (Lines 197, 198, 248, 249, 349, and 350).

  1. Please provide the total number of samples for each analysis. You provided the number of sites. Were single or multiple samples (replicates?) taken at each site?

Response: A total of 44 sites were sampled, with water samples collected from each site and sediment samples obtained from 25 of these sites. Therefore, the study comprised 44 water samples and 25 sediment samples in total. Those samples were used for all analyses. This information has been included in the Materials and Methods (lines 105-109, and 146). For each sample, no further replicates were prepared.

  1. The figures were very nice, but the similar colors made it a little difficult to see differences among the groups for Fig 2C. Also Fig. 2C needs more explanation in the legend. Fig. 4 provides an R value in the figure but it is not defined either in the methods or the legend.

Response: Thanks for rising the problem. As to increase readability, together with the color, abbreviations of each group were also added in the original Figure 2C. This helps to locate a specific group quickly. In this matter, we insist to keep the current Figure 2C. R2 values indicated the correlation coefficient. This has been added to the legends of Fig. 3 and Fig. 4 (lines 255, 256, and 315).

  1. The sampling occurred over a short period of time. It might be interesting to comment on how sampling at a different time or over a longer period might affect the results.

Response: We acknowledge that sampling at different times or over a more extended period of time could influence the results, including factors such as annual and seasonal variations. However, the primary objective of our study was to uncover spatial scale-dependencies in assembly processes. To mitigate temporal effects, we aimed to collect samples within a relatively short time frame. Given the absence of time-series samples in our study, we opted to concentrate on spatial effects and refrained from additional discussion on temporal influences.

Reviewer 2 Report

Comments and Suggestions for Authors

The original article " Different Microeukaryotic Trophic Groups Show Different Latitudinal Spatial Scale-Dependences in Assembly Processes across the Continental Shelves of China " by Yong Zhang, Zhishuai Qu, Kexin Zhang, Jiqiu Li and Xiaofeng Lin is a completed research paper. In this study, the authors investigated the latitudinal spatial-scale dependencies in the assembly processes of microeukaryotic communities in surface waters and bottom sediments along China's continental shelves. Undoubtedly, this work is of interest to a wide audience of readers.

All sections of the article are described clearly and concisely, the statement of aims and objectives of the study raises a relevant problem in a changing climate. During the review process I did not have any methodological issues, the Materials and Methods section is described as well as possible.  The work is based on extensive experimental data. The data have been analysed using modern statistical methods and support the conclusions made by the authors. The article is written in excellent scientific language, all sections are logically structured and form a coherent and understandable scientific work for the reader. The article has enough relevant literature references and excellent visuals.

I believe that this article can be accepted in its present form.

Author Response

The original article " Different Microeukaryotic Trophic Groups Show Different Latitudinal Spatial Scale-Dependences in Assembly Processes across the Continental Shelves of China " by Yong Zhang, Zhishuai Qu, Kexin Zhang, Jiqiu Li and Xiaofeng Lin is a completed research paper. In this study, the authors investigated the latitudinal spatial-scale dependencies in the assembly processes of microeukaryotic communities in surface waters and bottom sediments along China's continental shelves. Undoubtedly, this work is of interest to a wide audience of readers.

       All sections of the article are described clearly and concisely, the statement of aims and objectives of the study raises a relevant problem in a changing climate. During the review process I did not have any methodological issues, the Materials and Methods section is described as well as possible.  The work is based on extensive experimental data. The data have been analysed using modern statistical methods and support the conclusions made by the authors. The article is written in excellent scientific language, all sections are logically structured and form a coherent and understandable scientific work for the reader. The article has enough relevant literature references and excellent visuals.

       I believe that this article can be accepted in its present form.

Response: Many thanks for your time and effort for reviewing our manuscript.